# Retrospective Cohort Analysis of the Effect of Age on Lymph Node Harvest, Positivity, and Ratio in Colorectal Cancer

**DOI:** 10.3390/cancers14153817

**Published:** 2022-08-06

**Authors:** Samara L. Lewis, Kenneth E. Stewart, Tabitha Garwe, Zoona Sarwar, Katherine T. Morris

**Affiliations:** Department of Surgery, The University of Oklahoma Health Sciences Center, 800 Stanton L. Blvd, Oklahoma City, OK 73190, USA

**Keywords:** colon cancer, lymph node ratio, lymph node harvest, total positive lymph nodes, immune response

## Abstract

**Simple Summary:**

Colorectal cancer (CRC) is one of the leading causes of cancer-related death, with an increasing incidence in younger patients presenting with more aggressive tumor biology. One clinical marker of immune response that has been studied and found to correlate with overall survival and cancer-specific survival is the lymph node ratio. The goal of our study was to find if age alone, adjusting for known factors that impact nodal harvest, independently impacted lymph node ratio in CRC. We found that age does not impact lymph node ratio in a clinically relevant manner, but that lymph node ratio is strongly corelated with grade and histology of the tumor.

**Abstract:**

Introduction: Colon cancer among young patients has increased in incidence and mortality over the past decade. Our objective was to determine if age-related differences exist for total positive nodes (TPN), total lymph node harvest (TLH), and lymph node ratio (LNR). Material and Methods: A retrospective review of stage III surgically resected colorectal cancer patient data in the National Cancer Database (2004–2016) was performed, reviewing TPN, TLH, and LNR (TPN/TLH). Results: Unadjusted analyses suggested significantly higher levels of TLH and TPN (*p* < 0.0001) in younger patients, while LNR did not differ by age group. On adjusted analysis, TLH remained higher in younger patients (<35 years 1.56 (CI 95 1.54, 1.59)). The age-related effect was less pronounced for LNR (<35 years 1.16 (CI 95 1.13, 1.2)). Conclusion: Younger patients have increased TLH, even after adjusting for known confounders, while age does not have a strong independent impact on LNR.

## 1. Introduction

Over the past decade, the incidence of colorectal cancer (CRC) has steadily been increasing at a rate of 2% to 8% per year for patients younger than 50 years old [1,2,3,4,5,6]. Younger patients present at a higher stage, with higher grade, and with more aggressive histologic subtypes than those 60 years and older [7,8,9,10,11,12,13,14,15,16,17,18,19]. The reason for these clinical differences is not understood and needs further investigation. Recent studies have confirmed that the immune response to tumors has a significant impact on disease progression in CRC [20,21,22,23,24]. It is possible that age-related changes in the immune system could be contributing to the histological and clinical difference seen in these younger colorectal cancer patients. Age-related changes of the immune system such as senescence, or a decrease in immune cellular function could alter response to malignancy. Specifically, senescence results in declining immunosurveillance, which is necessary for cancer cell detection and elimination prior to disease presentation [25]. One clinical marker of the immune response to colorectal tumors that has direct correlation with overall survival is the lymph node ratio (LNR), defined as the number of total positive nodes (TPN) over the total number of nodes harvested (TLH) [26,27].

Previous studies have shown the number of total lymph nodes harvested (TLH) is impacted by age, with number of lymph nodes (LN) harvested declining with increasing age at diagnosis. These findings are independent of surgical specimen length and node positivity [28,29,30,31,32,33,34,35,36,37,38,39]. Whether this is due to decreased immune response secondary to age-related senescence is unknown. In addition, tumor size and stage of disease have a positive association with TLH [40]. Most suspect that LN hyperplasia contributes to LN identification within CRC specimens because larger LN facilitate identification by the pathologist during specimen evaluation [38,41,42]. A more robust immune response in younger patients could be contributing to increased TLH harvest seen in younger patients as increased LN hyperplasia would increase identification [43]. What is unknown is whether this age-related change in the immune response alters number of total positive nodes and the LNR.

The aim of this study was to identify if age at diagnosis is associated with significant differences in the total positive nodes and LNR of CRC patients, as well as determine if the association between TLH and age remains significant after controlling for stage, grade, histology, and tumor location. To accomplish our aim, we used the National Cancer Database (NCDB) to test the hypotheses that younger patients would have both increased LNR, independent of increased TPN as compared to their older counterparts. 

## 2. Methods

### 2.1. Source

This study utilized the National Cancer Database (NCDB), which is a joint project of the American College of Surgeons Commission on Cancer (CoC) and the American Cancer Society, capturing data on patients including demographics, socioeconomic factors, insurance status, tumor characteristics, and treatment details. These data are collected from >1500 CoC accredited hospitals. Trained personnel at each site collect data as outlined by the CoC’s Facility Oncology Registry Data Standards manual [44]. All patient information is de-identified and is released through a Participant User File (PUF), allowing for the study to be deemed exempt after evaluation by our Institutional Review Board. The project utilized the colorectal cancer PUF containing data from years 2004 to 2016.

### 2.2. Study Cohort

This study included stage III primary colorectal cancer patients who underwent surgical resection for which cancer-specific and lymph node information was reported within the NCDB. The effect of age was analyzed through dividing patients into 4 age cohorts. Patients were included if they had a diagnosis of primary colon or rectal cancer and underwent resection. All patients were limited to stage III patients so as to remove palliative and segmental resections limiting lymph node yield. Exclusion criteria included patients younger than 18 years old and histology confirming a malignancy other than CRC (e.g., ovarian cancer or lymphoma). As this study was limited to primary CRC, histology was further reduced to exclude any patients with carcinoid and neuroendocrine tumors. Histological codes were divided as such; adenocarcinoma (8330, 8144, 8140, 8210, 8323, 8290, 8310, 8261, 8260, 8211, 8213, 8220, 8221); mixed adenocarcinoma and carcinoma (8574, 8572, 8571, 8570, 8255, 8143, 8142, 8141, 8262, 8263, 8575, 8145, 8230, 8231); goblet cell (8243); mucinous and mucin producing (8470, 8481, 8471); and signet ring (8490). Location of the tumor within the colon was divided into four locations, right-sided tumors consisting of cecum, ascending colon, and hepatic flexure (C180, C182, C183); transverse including transverse colon (C184); left-sided tumors consisting of splenic flexure, descending colon, and sigmoid (C185, C186, C187); and rectum. Patients were divided into four age cohorts: less than 35 years old, 36 to 54 years old, 55 to 74 years old, and those 75 and older. Age groups were divided to capture the extremes of age, >35 and >74 years old, with the two middle-aged cohorts selected to include age ranges where national screening should have already begun to minimize lead time bias. Demographic data included sex, race, insurance status, and annual income. Primary outcomes included total positive lymph nodes (TPN), total lymph nodes harvested (TLH), and lymph node ratio (LNR), calculated by dividing total positive lymph nodes (TPN) by total lymph nodes harvested (TLH). Cancer-specific data collected included tumor behavior (invasive vs. in situ), grade, location of primary tumor, T stage, histology, microsatellite instability (MSI), and neoadjuvant chemotherapy.

### 2.3. Statistical Analysis

Bivariate associations of age group and categorical variables were assessed using chi-square tests. Differences in continuous variables across age groups were tested using analysis of variance (ANOVA). A significance level of alpha < 0.05 was used for all tests.

Patients with missing histology, stage, behavior, administration of neoadjuvant chemotherapy, and lymph node information, were excluded from multivariable analyses. TPN were analyzed only for patients that had >0 lymph nodes harvested. Covariates considered for multivariable adjustment included histology, behavior, grade, administration of neoadjuvant chemotherapy, pathologic T stage, and location of the primary tumor. Negative binomial regression with TLH as an offset variable was used to assess the relationship of age group and LNR while adjusting for other covariates. For this multivariable analysis, patients were included only if there was a minimum of 12 lymph nodes harvested with a maximum of 36. These cut points were used due to previously identified clinical significance and distribution of TLH across age groups. NCCN guidelines recommend a minimum of 12 lymph nodes for accurate staging and 36 lymph nodes have shown to be the upper limit of lymph node harvest to correlate with survival [45,46,47]. Additionally, our analysis of the distribution of TLH by age group indicated < 2% of TLH overlap across age groups beyond 36 lymph nodes. Adjusted parameter estimates with 95% confidence intervals were obtained.

The association of TLH with age group was estimated using multivariable negative binomial regression to obtain adjusted incidence rate ratios (IRR). All analyses were performed using SAS version 9.4 (SAS Institute, Cary, NC, USA). Due to clinical treatment differences that occur in rectal cancer versus colon cancer in both neoadjuvant treatment and surgical resection, a separate multivariable analysis of TLH and LNR was performed for colon cancer and rectal cancer to identify whether there was an impact on lymph node status that may not be accounted for by tumor location alone.

## 3. Results

A total of 272,270 stage III colorectal cancer patients were initially identified. After exclusion of patients with missing information for grade, histology, grade, and site 256,489 patients remained. Lastly, after exclusion of patients missing neoadjuvant treatment or pathological T stage, 215,471 patients remained for development of the multivariable models. The patient cohorts had a slight male predominance by age group (52.8%, 53.2%, 54.6%) except for those aged 75 and older (56.9% female) (Table 1). The majority of patients were of White race. The largest predominance of White race was in our greater than 75-year cohort with 88.8%. The majority of patients had some form of insurance. The youngest cohort, however, had the highest rate of uninsured at 9.6%. Average annual income distribution was similar across the age cohorts.

Adenocarcinoma was the most common histology seen in all age cohorts (82.6, 88.8, 88.3, 86.0%). Consistent with previous studies, the highest percentage of patients with signet ring histology was found in our youngest age cohort (5.3%). The predominant tumor location for our youngest age cohort was rectum (34.5%) while right-side tumors made up the majority in the eldest cohort (50.6% *p* < 0.0001). The most common T stage for all age groups was T3 at approximately 54% (53.1%, 54.6%, 55.2%, and 55.7%). There was a slightly higher frequency T4 stage in the youngest age group at 18.8%. Information regarding presence or absence of microsatellite instability (MSI) information was missing for nearly half of the patients; however, among those with data recorded, negative MSI predominated with range from 74.3% to 69.9% across the age cohorts (Table 2).

Mean TPN was found to be higher in the 2 youngest age cohorts while the two eldest were identical, with 4.8 average positive nodes in the patients < 35 years old, as compared to 3.9 in patients 36–55 years old, 3.6 in those aged 56–74, and 3.6 among those ≥ 75 years old (Figure 1). The mean TLH was highest among patients < 35 years old at 27.3 and decreased to 18.4 in the oldest age group (Figure 1). The mean LNR was similar across all age groups with a range of 0.21–0.23 (Figure 2).

After adjusting for tumor location, grade, histology, pathological T stage, and neoadjuvant chemotherapy the TLH was increased with decreasing age of the patient. The largest difference was noted when comparing the youngest cohort of patients (those ≤ 35) who had an incident rate ratio (IRR) 1.56 (95% CI 1.54, 1.59) times than that of patients age ≥ 75 (Table 3). The estimated LNR was only slightly higher among younger patients after adjusting for covariates, with the IRR in patients ≤ 35 of 1.16 (95% CI 1.13, 1.2) when compared to patients ≥ 75 (Table 3). Additional factors found to impact LNR were grade, histology, and pathologic T stage of the tumor. LNR directly increased as the grade of the tumor became more undifferentiated, with poorly differentiated tumors having the largest impact on LNR 1.60 times higher than well differentiated tumors (CI 95% 1.56, 1.62). More aggressive histologies were found to correlate with increased LNR, with signet ring having the largest impact, 1.56 times higher than in adenocarcinoma (CI 95% 1.52, 1.61). Pathologic T stage had the largest correlation with LNR with T4 tumors having 2.13 times increased LNR (Table 3).

When colon cancer was separated from rectal cancer, the impact of age was still noted in the TLH and TPN, with the youngest patients (≤35 years old) having the highest TPN and TLH (1.12 colon vs. 1.20, and (1.59 colon vs. 1.50 rectal). The impact of age on TPN and TLH were comparable between the colon and rectal as with each age cohort decreased in age, the rate of TPN and number of TLH increased in a similar fashion (Table 4).

## 4. Discussion

Nodal status in CRC is an important prognostic factor with overall survival inversely correlating with the LNR [48,49,50]. With the growing incidence of CRC in younger patients over the past decade the impact of age on these factors is critical to understand [1,2]. The patients represented in previous studies may not be reflective of the younger age patient population now being treated for CRC. The goal of this study was to identify if age alone impacted nodal status in colorectal cancer patients. Several studies have previously identified that TLH is greater in younger patients, but none of these studies adjusted for tumor specific factors that could impact nodal yield [38,40,41,49,51,52]. To our knowledge, this is the first study to confirm that younger age independently correlates with increased lymph nodes harvest regardless of tumor stage, location, grade, neoadjuvant chemotherapy or histology. This was also confirmed in separate analyses for both colon cancer and rectal cancer (IRR of 1.56 in patients ≤35 for all CRC and IRR 1.59 in colon and 1.50 in rectal).

Interestingly, age was not found to impact LNR as much as it affected TLH which could suggest that higher lymph node counts in younger patients are due to increased lymph node hyperplasia in younger patients leading to improved pathologic identification. Indeed, lymph node hyperplasia is known to aid in identification of lymph nodes within surgical specimens by pathologists, and these nodes appear to be enlarged consistently in younger patients independent of nodal metastasis. This could be a direct reflection of the age-related changes that exist in the immune system resulting in more robust response in younger patients.

Although, there was a statistically significant elevation in the LNR within our youngest age group (IRR 1.16), this did not cross the threshold for clinically significant effects on survival. Previously identified cut points for LNR found to negatively impact overall survival have been identified at 17%, 41%, and 69%, while within this study the mean LNR for all age groups was approximately 22%, indicating that age did not have a meaningful impact on LNR [49] (Figure 2). These results at first glance, contrast what has been previously reported, by Xei et al. and Meyer et al. who both found that younger age directly correlated with increased positivity of nodes utilizing the Surveillance, Epidemiology, and End Results database (SEER). Xei et al. in T1 and T2 colon cancer patients used a predictive model to identify those patients less than 40 years old were more likely to have an increased LNR and Meyers et al. utilizing Poisson regression models in stage I-III rectal cancer patients identified younger age and increased T stage correlated with increased node positivity [53,54]. Both of these studies did not adjust for the other known cancer-specific factors that can affect nodal positivity such as grade and histology. As shown within our study aggressive histology types such as signet ring have a significant association with nodal positivity (IRR 1.56 (CI 1.52,1.61) and grade of tumor also correlates with nodal positivity with poorly differentiated tumors having highest impact (IRR 1.60 (CI 1.56, 1.62). The results of these two prior studies in combination with our results, highlight the fact that younger patients are different from their elder counter parts and they do present with higher rates of positive nodes, but our study allows for the clarification that the major contributor to this finding, actually is from the nature of the tumor and not only the age of the patient.

The importance of the LNR has been previously established as it has a strong correlation with overall and cancer-specific survival [48,49,50]. It also serves as a clinical surrogate for the immune system interaction with the tumor and directly reflects the nature of the tumor, as shown within this study where histology, T stage, and grade correlated with LNR [38]. With the increasing incidence of younger patients with CRC, identifying that younger age has minimal clinical impact on LNR is important to allow for this biomarker to continue to serve as an accurate prognostic indicator.

## 5. Limitations

This study is limited by the retrospective nature of the data collection and, therefore, temporal and individual institutional alterations in treatment could not be accounted for and large percentage of missing data for MSI, an important variable. Additionally, due to the large sample size, *p*-value calculations, while indicating statistical significance, may not translate to differences that are clinically significant. As such, our discussion and conclusions have been limited to clinically relevant findings. This study is also limited to only CRC patients who underwent surgical resection and form of surgical intervention—open, robotic, or laparoscopic—was not accounted for due to the limitations of the data set.

## 6. Conclusions

Younger patients with colon and rectal cancer have higher lymph node harvest after adjusting for tumor location, histology, grade of tumor, and neoadjuvant chemotherapy. Younger age has a slight correlation with increased total positive nodes and lymph node ratio, but both these components of nodal status were found to be more largely impacted by tumors specific characteristics including grade, histology, and T stage.

## Figures and Tables

**Figure 1 cancers-14-03817-f001:**
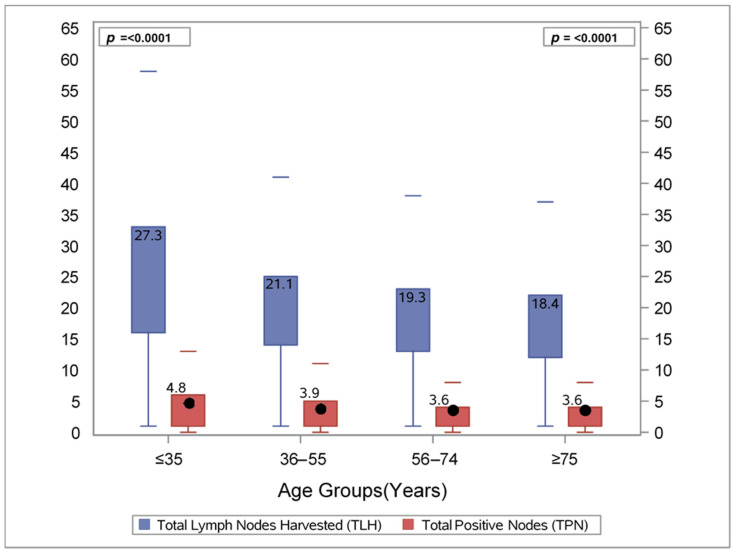
Mean total lymph nodes harvested (TLH) and total positive nodes (TPN) by age groups with at least one lymph node harvested.

**Figure 2 cancers-14-03817-f002:**
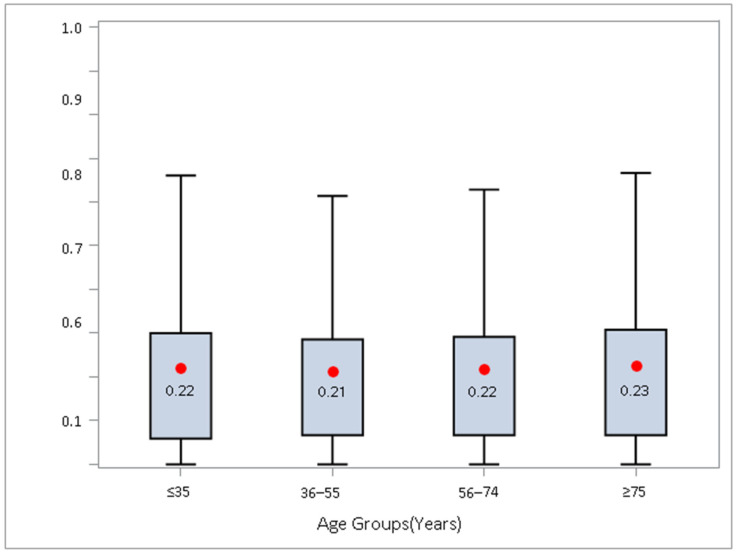
Mean LNR ratio by age groups with at least one lymph node examined.

**Table 1 cancers-14-03817-t001:** Sociodemographic factors by age groups (n = 272,270).

Variables	≤35 n = 5021 (1.8%)	36–55 n = 59,493 (21.9%)	56–74 n = 122,115 (44.9%)	≥75 n = 85,641 (31.5%)	*p*-Value
**Sex, n (%)**					
Male	2651 (52.8)	31,646 (53.2)	66,630 (54.6)	36,912 (43.1)	**<0.0001**
Female	2370 (47.2)	27,847 (46.8)	55,485 (45.4)	48,729 (56.9)	
**Race, n (%)**					
White	3944 (78.6)	46,543 (78.2)	100,212 (82.1)	76,061 (88.8)	**<0.0001**
Black	699 (13.9)	9009 (15.1)	15,296 (12.5)	6365 (7.4)	
Other	338 (6.7)	3397 (5.7)	5669 (4.6)	2661 (3.1)	
unknown	40 (0.8)	544 (0.9)	938 (0.8)	554 (0.7)	
**Insurance, n (%)**					
Not Insured	484 (9.6)	4399 (7.4)	4224 (3.5)	352 (0.4)	**<0.0001**
Privately Insured	3366 (67.0)	43,649 (73.4)	46,986 (38.5)	7611 (8.9)	
Medicaid/Medicare	981 (19.5)	9514 (16.0)	67,615 (55.4)	76,200 (89.0)	
Other Government	68 (1.4)	718 (1.2)	1271 (1.0)	390 (0.5)	
unknown	122 (2.4)	1213 (2.0)	2019 (1.7)	1088 (1.3)	
**Income, n (%)**					
<$38,000	920 (18.4)	10,914 (18.4)	23,768 (19.6)	13,951 (16.4)	**<0.0001**
$38,000–47,999	1203 (24.1)	13,311 (22.5)	29,554 (24.3)	20,386 (23.9)	
$48,000–62,999	1398 (28.0)	15,233 (25.7)	32,483 (26.7)	23,405 (27.5)	
≥$63,000	1474 (29.5)	197,25 (33.3)	35,683 (29.4)	27,506 (32.3)	
Income missing n = 1356 (0.5%) bolded indicate significance with *p* < 0.05					

**Table 2 cancers-14-03817-t002:** Cancer-specific properties by age groups.

Variables	≤35 n = 5021 (1.8%)	36–55 n = 59,493 (21.9%)	56–74 n = 122,115 (44.9%)	≥75 n = 85,641 (31.5%)	*p*-Value
**Histology, n (%)**					
Adenocarcinoma	4148 (82.6)	52801 (88.8)	107,882 (88.3)	73,637 (86.0)	**<0.0001**
Goblet Cell	6 (0.1)	73 (0.1)	123 (0.1)	39 (0.05)	
Mixed Adenocarcinoma and Carcinoma	2 (0.04)	62 (0.1)	140 (0.1)	118 (0.14)	
Mucin Producing	600 (12.0)	5445 (9.2)	11805 (9.7)	10146 (11.9)	
Signet Ring	265 (5.3)	1112 (1.9)	2165 (1.8)	1701 (2.0)	
**Location, n (%)**					
Rt Colon	1174 (23.4)	14,828 (24.9)	45,307 (37.1)	43,354 (50.6)	**<0.0001**
Transverse Colon	303 (6.0)	3050 (5.1)	8132 (6.7)	7763 (9.1)	
Lt Colon	1635 (32.6)	19,922 (33.5)	34,380 (28.2)	18,915 (22.1)	
Rectum	1733 (34.5)	20,386 (34.3)	31,380 (25.7)	13,339 (15.6)	
unknown	176 (3.5)	1307 (2.2)	2916 (2.4)	2270 (2.7)	
**T Stage, n (%)**					
T1	185 (3.68)	2752 (4.63)	4659 (3.82)	1860 (2.17)	**<0.0001**
T2	447 (8.9)	6192 (10.4)	11,692 (9.57)	6462 (7.55)	
T3	2667 (53.1)	32,503 (54.6)	67,369 (55.2)	47,695 (55.7)	
T4	943 (18.8)	8534 (14.3)	18,322 (15.0)	14,958 (17.5)	
Tx	764 (15.2)	9382 (15.8)	19,849 (16.3)	14,558 (17.0)	
**Microsatellite Instability, n (%)**					
Negative	965 (74.3)	9661 (81.7)	14,913 (78.8)	7793 (69.9)	**<0.0001**
Positive	334 (25.7)	2166 (18.3)	4010 (21.2)	3353 (30.1)	
MSI missing n = 229,075					

**Table 3 cancers-14-03817-t003:** Negative binomial regression, LNR and TLH.

Variables	LNR (IRR) (CI)	TLH (IRR) (CI)
**Age Cohort**		
≤35	1.16 (1.13, 1.2)	1.56 (1.54, 1.59)
36–55	1.09 (1.07, 1.10)	1.21 (1.20, 1.22)
56–74	1.03 (1.02, 1.04)	1.08 (1.07, 1.09)
≥75	Reference	Reference
**Neoadjuvant Chemotherapy**	0.89 (0.88, 0.91)	0.81 (0.80, 0.82)
**Tumor Location**		
Left	0.88 (0.87, 0.89)	0.99 (0.99, 1.01)
Right	0.88 (0.87, 0.89)	1.13 (1.13, 1.14)
Transverse	0.77 (0.75, 0.78)	1.11 (1.11, 1.13)
Rectum	Reference	Reference
**Grade**		
Well differentiated	Reference	Reference
Moderately differentiated	1.10 (1.08, 1.15)	1.01 (1.00, 1.01)
Poorly differentiated	1.60 (1.56, 1.62)	1.03 (1.02, 1.04)
Undifferentiated	1.56 (1.52, 1.61)	1.06 (1.04, 1.07)
**Histology**		
Adenocarcinoma	Reference	Reference
Goblet Cell	0.71 (1.04, 0.86)	0.84 (0.77, 0.92)
Mixed Adenocarcinoma and Carcinoma	1.35 (1.21, 1.52)	0.98 (0.92, 1.03)
Mucin Producing	1.13 (1.12, 1.15)	1.02 (1.01, 1.03)
Signet Ring	1.56 (1.52, 1.61)	0.98 (0.96, 0.99)
**Pathologic T stage**		
T2	1.24 (1.21, 1.28)	1.05 (1.04, 1.07)
T3	1.77 (1.72, 1.82)	1.12 (1.11, 1.14)
T4	2.13 (2.07, 2.19)	1.14 (1.13, 1.15)
Tx	1.91 (1.85, 1.97)	1.05 (1.04, 1.06)
**Total Lymph Node Harvested**	1.02 (1.01, 1.02)	

95% CI; TLH—Total Lymph Node Harvested, LNR—Lymph Node Ratio, IRR—Incident Rate Ratio, CI—Confidence Interval.

**Table 4 cancers-14-03817-t004:** Negative binomial regression, TLH and LNR colon vs. rectal cancer.

	Colon		Rectal	
Variables	LNR (IRR) (CI)	TLH (IRR) (CI)	LNR (IRR) (CI)	TLH (IRR) (CI)
**Age Cohort**				
≤35	1.14 (1.10, 1.19)	1.59 (1.56, 1.61)	1.20 (1.14, 1.28)	1.50 (1.46, 1.55)
36–55	1.09 (1.08, 1.11)	1.22 (1.22, 1.23)	1.07 (1.05, 1.10)	1.17 (1.56, 1.19)
56–74	1.03 (1.02, 1.04)	1.09 (1.08, 1.09)	1.01 (0.99, 1.04)	1.07 (1.05, 1.08)
≥75	Reference	Reference	Reference	Reference
**Neoadjuvant Chemotherapy**	1.02 (0.97, 1.07)	0.95 (0.93, 0.97)	0.87 (0.86, 0.89)	0.80 (0.80, 0.81)
**Grade**				
Well differentiated	Reference	Reference	Reference	Reference
Moderately differentiated	1.09 (1.07, 1.11)	1.01 (1.00, 1.02)	1.12 (1.08, 1.16)	1.01 (0.99, 1.02)
Poorly differentiated	1.59 (1.56, 1.62)	1.03 (1.02, 1.04)	1.60 (1.54, 1.67)	1.02 (1.00, 1.04)
Undifferentiated	1.55 (1.51, 1.60)	1.05 (1.04, 1.07)	1.59 (1.49, 1.70)	1.07 (1.03, 1.11)
**Histology**				
Adenocarcinoma	Reference	Reference	Reference	Reference
Goblet Cell	1.03 (0.85, 1.24)	0.85 (0.77, 0.92)		
Mixed Adenocarcinoma and Carcinoma	1.32 (1.17, 1.50)	0.97 (0.91, 1.03)	1.55 (1.12, 2.13)	1.01 (0.86, 1.18)
Mucin Producing	1.09 (1.08, 1.11)	1.02 (1.02, 1.03)	1.33 (1.29, 1.38)	1.01 (0.99, 1.03)
Signet Ring	1.51 (1.47, 1.56)	0.97 (0.95, 0.99)	1.82 (1.69, 1.97)	1.01 (0.97, 1.05)
**T Stage, n (%)**				
T1	Reference	Reference	Reference	Reference
T2	1.26 (1.21, 1.31)	1.06 (1.04, 1.08)	1.22 (1.16, 1.29)	1.05 (1.02, 1.07)
T3	1.79 (1.73, 1.85)	1.13 (1.12, 1.15)	1.74 (1.66, 1.83)	1.15 (1.08, 1.13)
T4	2.18 (2.11, 2.26)	1.14 (1.12, 1.15)	1.95 (1.85, 2.06)	1.17 (1.14, 1.20)
Tx	1.96 (1.90, 2.03)	1.06 (1.04, 1.07)	1.77 (1.68, 1.87)	1.03 (1.01, 1.06)
**Total Lymph Node Harvested**	1.02 (1.02, 1.02)		1.03 (1.03, 1.03)	

95% CI; TLH—Total Lymph Node Harvested, LNR—Lymph Node Ratio, IRR—Incident Rate Ratio, CI—Confidence Interval.

## Data Availability

The data presented in this study are available in this article.

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
