# Peer review of "Retrospective Cohort Analysis of the Effect of Age on Lymph Node Harvest, Positivity, and Ratio in Colorectal Cancer"

_cancers, 2022, doi:10.3390/cancers14153817_

Round 1
Reviewer 1 Report
Dear Authors,
This is a fine study with large numbers included.
INTRODUCTION
--well written. No comments.
METHODS
--no comments.
RESULTS
--abbreviations in the tables must be explained.
DISCUSSION
very short, no real discussion can be found.
Conclusions - I do not see the clinical relevance of this study. Younger patients have more lymph nodes removed compared to older. Should you somehow change the practice to increase the numbers? Should we still stick to the TLH? Search for the more precise staging factors?
Reviewer 2 Report
The authors performed a retrospective analysis of data from the NCDB with the objective to examine whether age-related differences regarding lymph node staging (total number, positivity, ratio) exist. They limited their analysis to patients with stage III CRC surgically treated within the years 2004 to 2016. The authors found that younger patients have a significantly higher LN harvest suggesting a pronounced immune response, a higher rate of positive LN but no clinically relevant difference in LN ratio.
The research question as well as the methodology seem sound and accurate, but with some limitations, as in my opinion relevant data and information are missing and the conclusion was somewhat oversimplified, as colon and rectal cancers were grouped together despite different treatment approaches and the emerging separation of the two pathologies. Nevertheless, I propose that the manuscript can be accepted for publication after some revisions and comments on the following points:
Please provide any other cancer-specific characteristics (Table 2). Most importantly, information on tumor histology (T stage) is missing in Table 2 and also in the regression analysis in Table 3, as the depth of invasion could also trigger an immune response and LN positivity, as you have explained yourself in the introduction (lines 52 and 62).
Another interesting analysis would be the overall comparison between colon and rectal cancer, including adjustments for neoadjuvant therapy.
The range of years included is quite wide and treatment approaches, including the recommended number of LNs removed, have changed over the years. Please provide information on whether there was an improvement in the total number of LNs in the later years compared to the earlier years or whether the timing of surgery had an impact on the LN ratio.
Please comment on the clinical relevance and implication of an age-dependent LN ratio.
Round 2
Reviewer 1 Report
Dear Authors,
The manuscript improved a lot.
Thank you
Reviewer 2 Report
I have read the revised mansucript with great interest. The additional analysis has greatly enhanced the value of the paper, and I would recommend that the mansucript be accepted for publication in its revised form.